# Volume of Fluid (VOF) Method as a Suitable Method for Studying Droplet Formation in a Microchannel

**DOI:** 10.3390/mi16070757

**Published:** 2025-06-27

**Authors:** Felipe Santos Paes da Silva, Paulo Noronha Lisboa-Filho

**Affiliations:** Department of Physics, School of Science, State University of São Paulo, Bauru 17033-360, SP, Brazil; fsp.silva@unesp.br

**Keywords:** microfluidics, microchannel, computational fluid dynamics, volume of fluid method, level set method

## Abstract

Microfluidics is a rapidly advancing field focused on optimizing microdevices for applications such as organ-on-a-chip systems and enhancing laboratory analyses. Understanding the physical parameters of droplet generation is crucial for these devices. Computational fluid dynamics (CFD) techniques are essential for providing insights into the limitations and efficiency of numerical methods for studying fluid dynamics and improving our understanding of various application conditions. However, the influence of different numerical methods on the analysis of physical parameters in problems involving droplet generation in microchannels remains an area of ongoing research. This study implements the Volume of Fluid (VOF) method to investigate key physical parameters, including droplet size and the effect of the capillary number on fluid regimes, in droplet generation within a microchannel featuring a T-junction geometry. We compare the VOF method with the widely used Level Set Method (LSM) to evaluate its suitability for this context. The results show that the VOF method agrees with the LSM in fundamental outcomes, such as the reduction in droplet diameter as the flow rate ratio increases and the identification of the capillary number’s influence on fluid regime classification. The VOF method provides a clearer understanding of transitions between fluid regimes by detecting stages of non-uniformity in droplet size. It identifies a transition region between regimes with variations in droplet size, proving to be effective at mapping fluid flow regimes. This study highlights the potential of the VOF method in offering more detailed insights into instabilities and transitions between fluid regimes at the microscale.

## 1. Introduction

The development of microdevices has played a significant role in biomedical applications, particularly those related to the optimization of lab-on-a-chip devices, which have become essential for adapting classical laboratory methods to the microscale volumes of biological phenomena [1].

Systems that integrate microdevices are founded on the principles of microfluidics, a technology focused on the manipulation of small volumes of fluids within microchannels typically sized between a few tens and several hundreds of micrometers [2]. The study of microfluidics applied to microdevice-based systems for biotechnological applications can focus on analyzing the physical parameters of the dynamics of biological fluids flowing through microchannels. This analysis enables the understanding of how fluid behavior, dictated by physical parameters, relates to and influences the biological parameters essential for optimizing a properly functional system.

Droplet microfluidics is classified as a subcategory of microfluidics that is of great interest, as it allows for the creation, manipulation, and analysis of tiny droplets, which can serve as independent microreactors for various chemical and biological reactions [3]. In this context, the ability to produce monodisperse microdroplets plays a significant role in devices used for organ-on-a-chip applications [4].

The implementation of computational tools emerges as an important means to facilitate the study of microfluidics in microdevices. Computational fluid dynamics (CFD) is characterized by the application of robust computational methods to model fluid flow conditions, predicting aspects of heat, mass, and momentum transfer [5] by employing numerical analyses and data structures to identify and solve fluid flow problems [6].

Computational fluid dynamics (CFD) techniques enable the determination of critical parameters such as fluid velocity, pressure, surface tension, and density, which are challenging to ascertain through experimental techniques. The analysis provided by computational modeling of microfluidic phenomena under a wide range of experimental conditions allows for the extraction of relevant information for the development and optimization of microdevices, as well as their operating conditions and functionality [7].

For droplet-based microfluidic systems, analyzing critical parameters becomes a fundamental task, as it is necessary to understand and identify their influence on the dimensions and shapes of the generated droplets, considering that monodisperse droplets with specific sizes are crucial for different applications [8]. Thus, precise control over the dimensions and shapes of the microdroplets generated in microchannels is important, as it influences critical parameters for the functioning of microdevices [9]. As for the analysis of microdroplet generation, mapping the flow regime is also an important task, as it helps to determine the best conditions for forming microdroplets with the desired properties [10].

Many studies have applied computational and experimental methods to analyze physical parameters in the droplet generation process within microdevices [11]. Experimental investigations have examined the influence of fluid elasticity on droplet formation in T-junction geometries, indicating that elastic properties in low-viscosity fluids promote the generation of secondary droplets with varying sizes [12]. Complementary numerical analyses have studied the formation of both single and compound droplets in microfluidic flow-focusing devices, utilizing finite element methods with adaptive mesh refinement to evaluate viscoelastic effects on droplet behavior [13]. Furthermore, the dynamics of droplet formation in flow-focusing configurations were predicted using the Volume of Fluid (VOF) method, revealing qualitative agreement between simulations and experimental observations [14].

It is observed in the literature that many studies have implemented different computational methods to evaluate the physical properties of droplet formation in microchannels. However, few efforts have been made to comparatively assess the efficiency of different methods and to understand the similarities and discrepancies between them. Thus, the present work aims to understand the main differences between two well-established computational methods within the context of microfluidics through the study of droplet formation in a microchannel with T-junction geometry.

Building upon this foundation, we aim to analyze the effects of fluid velocity and capillary number (Ca) on the formation of microdroplets in a T-junction geometry channel using the Volume of Fluid (VOF) method. This includes identifying the influence of these parameters on droplet dimensions and their shapes, as well as examining the fluid regimes that dominate microdroplet formation. Alongside this, we seek to compare the VOF method with the Level Set Method (LSM) employed in the numerical–experimental study that served as our validation benchmark [11].

The distinct methods analyzed in this study exhibit unique potentialities and characteristics. While many studies highlight the Level Set Method as an excellent tool for analyzing droplet generation in microchannels due to its high-order accuracy and good mass conservation, we aim to explore how well the Volume of Fluid method adapts to microfluidic problems. By focusing on its potential for mass conservation, we hope to better understand phenomena and disturbances in the droplet generation process, particularly under conditions of fluid flow instability and variations in the capillary number (Ca).

## 2. Mechanisms of Droplet Generation

The formation of droplets in microchannels occurs due to the instability of the fluids present. Droplet generation typically occurs within three main regimes, known as squeezing, dripping, and jetting [15].

Microdroplets can be formed in a geometry that features a T-junction (Figure 1), where two perpendicular channels intersect. The main channel contains a continuous phase fluid, and the orthogonal channel is the inlet of a internal phase. At the beginning of droplet formation, the dispersed phase fluid flows into the main channel. In the following stage, this fluid forms a plug whose length is approximately the same as the width of the main channel. At this stage, there is an interaction between shear forces and interfacial forces.

The effect of interfacial tension and viscosity acting between two considered immiscible fluids is represented by the capillary number (Ca), which thus influences the conditions that determine the dominance of each regime in the droplet formation process, making it a determining factor in the characterization and control of droplet formation dynamics. This parameter can then be determined as(1)$$Ca=\frac{\mu U_{c}}{\sigma }$$
where *U* (m/s) is the average velocity of the carrier fluid, μ (Pa·s) is the dynamic viscosity, and σ (N/m) is the surface tension. The squeezing and dripping mechanisms typically occur due to the influence of the physical confinement exerted by the microchannel walls. In the squeezing regime, observed for values of $Ca<10^{-2}$, surface forces dominate over shear stresses, causing the drop length to depend on the flow rate ratio of the fluids and the dimensions of the microchannel. The transition between the two regimes occurs for values of $Ca>10^{-2}$, where, in this case, the dripping regime becomes dominant, characterized by the shear forces overtaking the surface forces [16].

In the squeezing regime, the flow rate of the fluid that constitutes the dispersed phase ($Q_{d}$) is greater than the flow rate of the continuous phase fluid ($Q_{c}$), and thus, the droplet elongates along the main microchannel, presenting a part of its shape that tapers. The breakup of the droplet occurs with the detachment of the dispersed phase from the channel junction due to the pressure drop over the droplet in the main channel. In the dripping regime, the flow rate of the dispersed phase fluid ($Q_{d}$) is lower than the flow rate of the continuous phase fluid ($Q_{c}$). Thus, the breakup of the droplet occurs immediately after a portion of the dispersed fluid is formed that extends completely over the cross-section of the main channel, creating a kind of plug. For high Ca values, shear forces become even more crucial. The continuous phase applies a shear stress on the growing droplet, causing it to detach near the junction at the upper wall, while still allowing the main channel’s cross-section to remain unblocked. This phenomenon is referred to as a shearing regime. Once detached, the droplet travels downstream within the main channel. Concurrently, the tip of the dispersed phase retracts to the end of the inlet channel and the cycle recommences. Therefore, the droplet size formed in the microchannel is governed by the balance between shear force and interfacial tension [11].

In this study, using computational fluid dynamics techniques, we employed a T-junction geometry to lead an analysis of properties such as the size and shape of microdroplets generated by the interaction between two fluids, oil (the continuous phase) and water (the dispersed phase), as well as a mapping of the flow regimes in which the microdroplets were generated. We considered $U_{d}$ as the velocity of the dispersed phase fluid and $U_{c}$ as the velocity of the continuous phase fluid. The flows (oil as continuous phase and water as dispersed phase) are considered two-dimensional and laminar. Both phases are considered to have constant fluid density and viscosity, and the capillary number (Ca) was determined based on the properties of the continuous phase.

## 3. Computational Method and Numerical Solution

Different methods can be used for numerical simulations of droplet formation under conditions of strong surface tension effects [17]. These methods can be classified as interface tracking or interface capture methods. Among the main methods applied are the Level Set Method, marker methods, the Phase Field method, and the VOF method, which can be combined with spatial and temporal discretization approaches, such as boundary integral methods, finite element methods, and finite difference schemes [18].

In general, computational methods for solving numerical simulations in fluid dynamics can be categorized into explicit interface tracking methods and implicit interface capture methods. The first group of methods involve tracking the interface using a moving grid or through the use of massless markers positioned along the interface. In addition, a moving grid coupled with a fixed grid can also be used. Among these methods, the Front Tracking method stands out [19].

Explicit methods reconstruct the interface by relying on a set of discrete points stored along the interface, with the resolution directly depending on the number of such points. Conversely, implicit methods employ a scalar function defined on a fixed grid to approximate the interface, where its position is inferred through an indicator function. As a result, these approaches do not provides the exact location of the interface.

In problems of dimensions *n*, explicit methods utilize a grid of dimensions (n−1) to reconstruct the interface, whereas implicit methods require solving an equation of dimensions *n* throughout the computational domain. A notable advantage of the implicit approach is its ability to address more complex problems, including those involving topological changes. Prominent examples of implicit methods include the Volume of Fluid (VOF) method, the Phase Field method, the Lattice Boltzmann method, and the Level Set Method.

A brief comparison of the main methods used in multiphase fluid dynamics simulations is presented below. Particular attention is given to the Volume of Fluid (VOF) method, as it was chosen for use in this study.

### 3.1. Front Tracking Method

The Front Tracking method, classified as an explicit interface tracking method, was developed by the Glimm and Tryggvason groups, who developed a three-dimensional Front Tracking algorithm to address Rayleigh–Taylor instability issues [20]. In its development, Unverdi and Tryggvason also investigated the rising of single and multiple bubbles [21]. This method employs interconnected markers to dynamically reconstruct the interface grid, establishing a relationship where the front grid moves relative to the fixed global grid. This approach provides an accurate representation of the location and geometry of the interface. In this context, the interfacial tension is calculated on the front grid and subsequently transferred to the fixed global grid.

Assuming that the physical properties of the fluid are constant within each phase, the Navier–Stokes equations are computed using a fixed grid. A discrete delta function, applied along the smoothed interface, is used to incorporate the interfacial jump condition into the momentum equations. This multigrid scheme demonstrates the complexity inherent in the Front Tracking method. Furthermore, it lacks the ability to resolve the formation of thin films resulting from droplet rupture [19].

The Front Tracking method has been improved and developed to address volume conservation and multiphase problems [22]. Some studies have implemented the Front Tracking method to demonstrate that discontinuity curves in two-dimensional interfaces can be tracked in a one-dimensional manner [23]. Furthermore, the algorithm has been uniformly applied to *n*-dimensional problems. Researchers have sought to develop Front Tracking methods to handle unconnected points in meshes [24].

### 3.2. Level Set Method

The Level Set Method, classified as an implicit interface capturing method, was demonstrated through a novel algorithm by Osher and Sethian, who developed a specific methodology for tracking a moving interface within a fixed grid system [25]. The Level Set function $\phi (\vec{x},t)$ is defined throughout the domain near the interface and represents the signed minimum distance from any point to the interface, measured along the normal direction. The interface between two fluids is implicitly represented by the function $\phi (\vec{x},t)=0$. This means that the interface is described as the set of points where the Level Set function equals zero, allowing for a smoother and more continuous representation of interfaces, even in the presence of deformation. For values of $\phi (\vec{x},t)>0$, the computational domain is referenced as being inside the interface—this is, on one side of the fluid. Conversely, for values of $\phi (\vec{x},t)<0$, the domain is considered to be outside the interface [26].

The method has been developed over the years, with significant contributions made by Olsson and Kreiss [27], who utilized it for interface capturing. One advantage of this method is that numerical calculations can be performed on a fixed Cartesian grid without the neeed for parameterizing the interface [7]. Due to its strong performance in accurately delineating phase boundaries, the Level Set Method has been widely applied to problems involving moving interfaces and free boundaries within the context of microfluidics [11]. The method has also been implemented in two-dimensional and three-dimensional problems, such as particle encapsulation in droplets transported by fluid flow [28]; droplet solidification and spreading [29]; droplet falling down [30]; breakup into smaller droplets [31]; and topological changes in the interface [32].

### 3.3. Volume of Fluid Method

The Volume of Fluid (VOF) method, classified as an implicit interface capturing method, was developed and extensively discussed by researchers such as Scardovelli and Zaleski [33], as well as Hirt and Nichols [34]. These authors presented techniques for employing the VOF approach to capture interfaces within non-uniform mashes. The method is based on a volume fraction function, which is then defined as a volume fraction field across the entire computational domain. The VOF method has been widely used in areas beyond multiphase fluid problems, such as heat transfer [35], thermocapillary motion [36], mass transfer [37], and particle-laden flows [38].

In this study, we guided the computational fluid dynamics simulations using the VOF method. This method characterizes the interface by using the volume fraction of one fluid phase or component, represented as *C*. In this method, the volume fraction function is introduced and defined within a volume fraction field across the entire computational domain [39]. In the bulk phase, within multi-fluid computational cells, the volume fraction C satisfies 0<C<1. Generally, the VOF model utilizes phase averaging to estimate the proportion of continuous and dispersed phases within each computational cell [19].

A variable α is introduced such that α=1 indicates a cell fully occupied by the continuous phase; α=0 corresponds to a cell entirely filled with the dispersed phase; and values of α between 0 and 1 denote cells containing the interface between both phases. The function of the volume fraction is (2)$$\frac{\partial \alpha }{\partial t}+\vec{U}\cdot \nabla \alpha =0$$
and can be solved on a fixed grid. In order to solve the function, the approximate position of the interface needs to be found. Additionally, the interface reconstruction must be carried out to determine the weighted viscosity and density for the computational cells, thus making it possible to calculate the volume flux for the convective terms present in the governing equations [19].

The density ρ and viscosity μ for the water and oil phases are determined based on a linear dependence [34]. For the continuous liquid (primary) phase (3) and dispersed phase (droplet) (4), it is described as(3)$$\rho =\rho_{1}\alpha +\rho_{2}\left(1-\alpha \right)$$(4)$$\mu =\mu_{1}\alpha +\mu_{2}\left(1-\alpha \right)$$

### 3.4. Governing Equations

The governing equations stem from the Navier–Stokes equation, which represents a continuous perspective of Newton’s second law per unit of volume. For a laminar and incompressive flow, it can be written as(5)$$\rho \left(\frac{\partial \vec{U}}{\partial t}+\vec{U}\cdot \nabla \vec{U}\right)=-\nabla p+\mu \nabla^{2}\vec{U}+\vec{f}$$

Additionally, the continuity equation is also solved:(6)$$\nabla \cdot \vec{U}=0$$

In the equations above, ρ is the density (${\mathrm{kg}/m}^{3}$), $\vec{U}$ is the velocity (m/s), *p* is the pressure (Pa), so ∇p represents its gradient, μ is the viscosity (Pa·s), and $\vec{f}$ represents the external forces applied to the fluid (${N/m}^{3}$).

### 3.5. Numerical Calculation of Surface Tension Force

Considering the Navier–Stokes formulation (Equation (Equation 5)), the term $\vec{f}$ incorporates the surface tension forces, which we denote as ${\vec{f}}^{\sigma }$. The surface tension force can be considered per unit volume. Numerical approximations for calculating surface tension forces have been developed over the years [40]. For the purposes of this article, we adopt the Continuous Surface Force (CSF) model, implemented through the resolution software used for the simulations.

Considering an elementary volume (Ω) intersected by two points *A* and *B*, the surface tension force is given by(7)$$\int_{\Omega }{\vec{f}}^{\sigma }\;dS=\oint_{A}^{B}\sigma \;d\vec{t}$$
where σ is the surface tension and $\vec{t}$ is the unit tangent vector. Using the Frenet parameterization for curves, we have $d\vec{t}=\kappa \vec{n}\;ds$, where κ is the curvature, $\vec{n}$ is the unit normal vector, and *s* is the curvilinear coordinate. Thus, we obtain the following:(8)$$\int_{\Omega }{\vec{f}}^{\sigma }\;dS=\oint_{A}^{B}\sigma \kappa \vec{n}\;ds=\int_{\Omega }\sigma \kappa \vec{n}\delta_{s}$$
where $\delta_{S}$ is the surface Dirac delta function, which is nonzero only at the interface.

Using the Continuous Surface Force model by Brackbill et al. [41], we can write the following relation:(9)$$\sigma \kappa \vec{n}\delta_{s}=\sigma \kappa \nabla H\left(\vec{x}-\vec{x_{s}}\right)$$
where *H* denotes the Heaviside function and $x_{s}$ represents the interface location, offering a smoothed Dirac delta approximation of the surface tension force. In the CSF model, a void fraction field $H_{\epsilon }=C$ is employed to approximate the Heaviside function, with ϵ=Δ corresponding to the grid resolution, such that(10)$$\lim_{\epsilon \to 0}H_{\epsilon }=H$$

The curvature of the interface can be determined using the height function approach, proposed by Cummins et al. [42] and Popinet [43], which is based on the concept that an interface can be described and defined as a graph of a function in a local coordinate system. The process consists of three steps: determining the local orientation of the coordinates through the vector component $\vec{n}$; estimating the height function by summing the void fraction values; and finally, calculating the curvature using central differencing [40].

Starting from the principle that the present work seeks its analyses through two-dimensional simulations, some considerations must be taken into account. The surface tension should be defined along a line, rather than a surface, with the normal vector $\vec{n}$ and the tangent vector $\vec{t}$ adjusted in a two-dimensional plane. Furthermore, the surface tension force can be considered along the interface line, which can be represented as(11)$$\int_{\Gamma }\sigma \;dL$$
where Γ is the interface line. The height function then describes the position of the interface line in one direction, and the curvature estimation can be made based on the orientation of the interface in a single plane. The Dirac delta function δ will also act along the interface line.

## 4. Computational Procedure and Simulational Setup

For this study, the CFD software Fluent was employed, which utilizes control-volume-based methods to discretize the governing equations into algebraic forms suitable for numerical solution. The geometry modeling used in the simulations was carried out using Ansys SpaceClaim (2024 R2) software and the mesh generation was carried out afterward, as shown in Figure 2. The quadrilateral mesh was used, providing more accurate calculations of surface tension effects compared to triangular or tetrahedral meshes [44]. The dimensions of the channels can be seen in Figure 3.

### 4.1. Mesh Independece Study

To ensure the adequacy of the number of nodes in the droplets and computational performance, we conducted a study on mesh independence. To this end, we performed a series of simulations with different mesh element sizes, analyzing how the mesh dimensions would alter or influence the droplet size parameters. The mesh independence study aims to identify the mesh element size from which variations in physical parameters (such as droplet size, in this case) become minimized despite further changes in element size. This makes it possible to employ a computational mesh that provides good accuracy while maintaining a reasonable computational cost. Thus, keeping all other variables fixed, we conducted simulations using different mesh element sizes.

To this end, we chose to test meshes with element sizes of 2.5 μm, 5.0 μm, 7.5 μm, and 10 μm in order to measure the computational effort. Figure 4 shows the different results for the droplet diameter for the various mesh element sizes. The analysis allowed us to consider the mesh size of 5 μm as the most efficient, as it provided considerably good accuracy while maintaining a reasonable computational effort.

The analysis of the results presented in the graph allows us to conclude that the mesh element size can directly influence the studied physical parameter—in this case, the droplet size. Although smaller mesh sizes, such as 2.5 μm, provide excellent precision in defining the interface, allowing an accurate estimation of droplet size, such mesh configurations require significant computational effort. However, larger meshes, such as those beyond 7.5 μm, reduce computational cost at the expense of the quality and definition of the fluid interface, thus reducing the accuracy of the simulation results. Therefore, we selected the 5 μm mesh, as it demonstrated an optimal balance, offering reliable simulation results with negligible differences compared to more refined meshes while maintaining a reasonable computational effort.

### 4.2. Simulational Setup

The validity of our model was established through comparison with the results of Bashir et al. (2011) [11]. For this validation, the computational setup parameters and fluid physical properties were kept consistent with those utilized in that study. To perform the simulation, we defined the initial conditions of phase interactions based on anti-diffusion interfacial modeling, assuming the surface tension coefficient to be constant and equal to 0.005 N/m. The inlet conditions were derived from the initial velocities of the fluids. A no-slip condition was applied on the walls of the geometry, and the contact angle between the phases, which constitutes the wall adhesion, was set to 135°. The solution method was characterized by the coupled with volume fractions scheme for the pressure–velocity coupling scheme. The spatial discretization was carried out using the Green–Gauss Cell-Based Gradient method, while PRESTO! was used for the pressure discretization scheme. The second-order upwind scheme was used for the discretization of the momentum equation. The simulations reached convergence when the time step accuracy was within $10^{-5}$, allowing for the creation of a database.

For all the simulations performed, we characterized the fluids with the physical properties presented in Table 1.

In order to understand the behavior of the fluids with respect to the generation and formation of microdroplets, we conducted simulations by varying the velocity of the continuous phase fluid ($U_{c}$) so that the ratio $U_{c}/U_{d}$ ranged from 1 to 12, considering the velocity of the dispersed phase fluid ($U_{d}$) fixed at $U_{d}=0.012$ m/s. Thus, we also varied the capillary number within this range. These values were chosen to ensure the compatibility of the present study with the work of Bashir et al., which served as a reference for data comparison [11].

## 5. Model Validation

As presented, considering the existence of different methods and simulation implementation approaches for the analysis of fluid dynamics behavior in microdevices, in this study, we aim to compare and validate our analyses through the work of Bashir et al., who implemented the Level Set Method (LSM), which is distinct method from the one used in the present study. Thus, we seek to identify the discrepancies resulting from the differences between the approaches, aiming for a more specific analysis of the Volume of Fluid (VOF) method’s implementation. In this sense, we propose an analysis of how the method employed in this study prioritizes physical parameters and under which conditions it differs from the Level Set Method, which was experimentally validated through the work of Bashir et al. [11].

Bashir et al. [11] presented an experimental investigation of droplet generation in water and oil phases using a T-junction geometry, with the same dimensions for the lateral and main channels as those used in our study. They identified a certain divergence between the sizes of droplets generated experimentally and those analyzed through simulations at low velocity ratio values ($U_{c}/U_{d}$). However, for higher flow rates, this difference diminishes. This discrepancy may be due to variations in the experimental physical properties. Moreover, performing 2D simulations tends to fail in accurately capturing the forces of tension acting on the droplet. Figure 5 shows a graphical comparison of the relationship between the dimensionless droplet length, given by the ratio of its diameter ($L_{d}$) to the channel width ($W_{c}$) in which the microdroplets are generated, and the velocity ratio ($U_{c}/U_{d}$).

In this study, we conducted computational simulations considering a viscosity ratio $\eta_{d}/\eta_{c}$ close to 0.8 so that the results obtained could be compared with the experimental model provided by the literature, thus validating the analyses carried out. It can be observed that, for low fluid flow rate values, there is a certain divergence in the results (Figure 3), a phenomenon also reported by Bashir et al. As the velocity ratio values increase, the numerical data obtained through the simulations in this study converge, to some extent, with the experimental data of the model on which this study is based. We can attribute the observed differences in droplet size behavior to the numerical methods used. While the Level Set Method, employed by Bashir et al., uses a distance function to define the interface, the Volume of Fluid (VOF) method uses a volume fraction function to represent the interface between phases. In this way, VOF is considered a more robust method in terms of mass conservation, while the Level Set Method may offer better accuracy in the shape of the interface but introduces errors with respect to mass conservation [45].

## 6. Results and Discussion

The analyses and subsequent discussions aim to emphasize the physical aspects of the droplet generation process in the studied microchannel by comparing the implementation of the Volume of Fluid (VOF) method with the numerical results obtained by Bashir et al. using the Level Set Method (LSM) [11]. As will be identified, the method implemented in this study enabled a specific analysis of regions of unstable droplet formation, which allowed the development of a more complete phase diagram of the fluid regime, also considering the transition zones.

For a T-junction microfluidic channel geometry, a pressure gradient produced by the shear stress forces of the continuous phase is responsible for the behavior of the dispersed fluid, which tends to break at the junction of the geometry and eventually forms a droplet [46]. The ways in which the droplet is generated, as well as its behavior along the channel, can be influenced by parameters such as fluid velocities, influencing the fluid flow rate ratio, as well as the viscosity ratio of the fluids. The following results show the effects on droplet formation, with respect to their sizes and fluid flow regimes, by altering the fluid flow rates.

### 6.1. Effects of Flow Velocity Ratio on Droplet Generation

A series of twelve simulations were conducted to validate our analyses on the formation of microdroplets in terms of their size through the presented experimental model in order to analyze the results obtained from the implementation in our simulation model. Thus, we varied the fluid velocity ratio from 1 to 12, keeping the velocity of the dispersed phase fluid (water) constant at $U_{d}=0.012\;m/s$ and altering the velocity of the continuous phase fluid (n-dodecane oil). The correlation between the formed droplet size and the velocity ratio is shown in Figure 6.

As can be observed, in general, the droplet size, regarding its diameter dimensions, decreases as the velocity ratio of the fluids increases. Similarly to the results presented by Bashir et al., the range of difference between the analyzed sizes decreases at high velocity ratios, whereas for low values, the differences become more significant. It is noted that the discrepancies observed when comparing our results with those from the numerical validation model occur during moments of fluid regime transition, as will be explained in the analysis of the capillary number. This can be explained by the fact that the method used in our analyses tends to show better performance in conserving the mass of the simulated system [45].

The discrepancies found, particularly under low-velocity-ratio conditions, are related to fluid regime transition moments, which explain a certain irregularity in droplet size. During transitions between regimes, a significant variation in droplet size may occur, as there is greater instability in the control over droplet separation, contributing to a broader distribution of droplet sizes. In these transitions, a combination of forces begins to act in a variable manner, further corroborating the irregularity [47].

Figure 7 shows the dependence of fluid regimes on the fluid velocities. We observe three regimes being formed. The first regime, observed for relatively low velocity ratios, is the squeezing regime, which has the pressure force as the cause of droplet breakup; the generated droplets are large and often irregular in size due to the longer detachment time and higher interfacial tension values [48]. The second regime observed is the dripping regime, in which a droplet is formed near the junction between the channels of the geometry; the velocity of the dispersed phase fluid is fast enough that the viscous stress exerted by the continuous phase fluid exceeds the interfacial tension. Thus, to comply with the mass conservation principle, the liquid thread begins to thin at the point of highest velocity. In this situation, a high capillary pressure is generated at the end of the fluid stretching, breaking the dispersed phase and, consequently, forming a droplet. The third regime observed, more subtly, is the jetting regime, which occurs with the increase in the velocity of the continuous phase fluid. The droplet tends to be formed at the end of a jet of the dispersed phase fluid due to the high flow inertia of this phase, causing the breakup to occur due to the instability of the jet [39]. A transition region from the squeezing to the dripping regime can also be observed. In this region, the droplets are not formed uniformly and do not tend to display a regular breakup mode.

### 6.2. Effects of Capillary Number on Droplet Size

By altering the fluid flow ratio through changes in the continuous fluid velocity ($U_{c}$), we also modify the capillary number for each simulation. From this, we could observe the behavior of the fluid phases with respect to droplet formation. Initially, as shown in Figure 8, it can be observed that the droplet size decreases as the capillary number increases, similar to what is observed with the fluid velocity ratios, since, under our conditions, the only factor that alters the capillary number is the variation in the velocity of the continuous fluid.

To conduct a more detailed analysis, we identified the moment of breakup that generated the droplet in the main microchannel The visual analysis of droplet breakup and formation enables a classification of the identified regimes according to the associated capillary number. Figure 9 shows the breakup process and the subsequent formation of droplets in the main channel concerning the capillary number (Ca) for each of the twelve simulations performed.

For relatively low capillary number values (Ca≤ 0.012), the presence of the squeezing regime is identified, which, as explained, results in the formation of larger and wider droplets, often without a well-defined contour. The time for the detachment of droplets is longer and there are irregularities between the interface of fluids, often producing droplets with poorly and non-uniform sizes [49]. For capillary number values within the range of 0.0012<Ca≤0.023, a transition stage between the regimes is observed. The droplets are formed irregularly, and the moments of droplet breakup exhibit different characteristics. In these transition regions, the pressure drop reaches a critical value, causing the transition from the squeezing regime to the dripping regime. It is also observed that, in this region, the droplet size does not continue to show a local decreasing trend; there is a slight increase in its diameter before returning to the trend of decrease. For capillary number values 0.023<Ca≤0.032, we observe droplet formation through the dripping regime, where droplets are generated near the junction of the geometry. In this regime, the role of shear force is significant for breakup. The droplets are typically more monodisperse and homogeneous compared to the squeezing regime and the transition phase [50]. Finally, for relatively high capillary number values (Ca>0.032), we identify the jetting regime. In this regime, droplets can be produced due to a Rayleigh–Plateau instability (i.e., capillary instability). The droplets are produced at the end of an extension of the dispersed phase fluid [51,52].

The mechanisms of droplet breakup and formation in the dripping and jetting regimes are different, influencing the shape and size of the generated droplets. In the dripping regime, the dispersed phase is influenced by several forces, including pressure, interfacial tension, viscous forces, and inertia. At the junction, the interfacial tension opposes droplet detachment, whereas pressure and viscous forces drive the dispersed phase downstream. These forces act on three distinct levels: initially, the dispersed phase enters the junction, inducing a pressure drop and velocity gradient in the continuous phase; subsequently, the droplet elongates within the main channel, but its growth is restrained by interfacial tension; and finally, when the dispersed phase obstructs the junction, the pressure drop reaches a critical threshold, leading to thinning of the elongated droplet. The breakup then occurs after the detachment of the dispersed phase from the junction. Thus, droplets are formed near the junction and attain their stable shape [53]. The jetting regime produces even smaller droplets due to the narrower elongation of the dispersed phase and lower values of interfacial tension and shear [54].

Figure 10 allows the visualization of different flow regimes in which fluids act during droplet generation, relating them to the variation in capillary number. It is possible to visually detect and classify, according to the capillary number and fluid flow rate of each regime, the regions where each type of breakup and droplet generation predominates. It can be observed that in the squeezing regime, the droplet size decreases as the capillary number increases. After reaching a certain value of Ca, there is a transition between regimes, and at this stage, the droplet size no longer follows a uniform trend of decrease or increase. Upon entering the dripping regime, the droplet size decreases as the capillary number increases. This decreasing trend continues into the jetting regime, but with a progressively slower rate of decrease.

Based on the analyses enabled by the results obtained in this study, it is possible to comparatively establish qualitative characteristics of the different evaluated methods, namely the VOF method, which was the primary approach employed in this research, and the LSM method, widely used in other studies and serving as numerical validation and comparison for this work. Table 2 provides a comparative description of the characteristics of the analyzed methods.

## 7. Conclusions

In this study, we aimed to understand how physical parameters, such as fluid velocity ratio and capillary number, affected the formation and generation of microdroplets in microchannels. To this end, we employed a numerical and simulation-based method that, once validated against literature data, provided crucial information for understanding the conditions of droplet formation. Initially, we analyzed how the velocity ratio between the dispersed and continuous phases ($U_{c}/U_{d}$) influenced the droplet size. We observed an inverse relationship, where droplet diameter decreased as the velocity ratio increased. Additionally, we examined the influence of the capillary number on the characteristics of different formation regimes. The employed simulation method, the Volume of Fluid (VOF) method, proved effective in identifying a transition region between formation regimes, characterized by instability in droplet sizes. The approach enabled a more detailed description of the mapping of fluid regimes.

Despite not offering the same level of accuracy in resolving the fluid interface as the Level Set Method, the VOF method demonstrated efficiency and proved to be suitable for analyzing microfluidic problems. It emerged as a viable alternative for numerical simulations in this field, balancing computational cost and accuracy. Moreover, the method was efficient in presenting key physical parameters, such as droplet size and formation regime characteristics, which are crucial for understanding the droplet generation process. The information obtained from this study contributes to a better understanding of the implementation of the VOF method in the numerical and simulation-based analysis of droplet formation in microchannels, with potential future applications in the development of microdevices.

## Figures and Tables

**Figure 1 micromachines-16-00757-f001:**
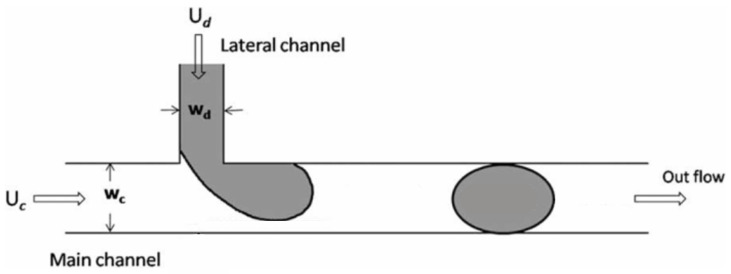
Schematic of the t-junction microchannel. A lateral channel is used for the inlet of the dispersed phase, while the continuous phase flows through the main channel. At the junction, the two phases converge, initiating the droplet formation process. The arrows in the schematic indicate the direction of the flows.

**Figure 2 micromachines-16-00757-f002:**
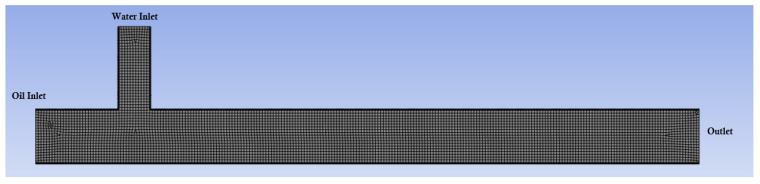
Meshing applied to the modeled geometry.

**Figure 3 micromachines-16-00757-f003:**
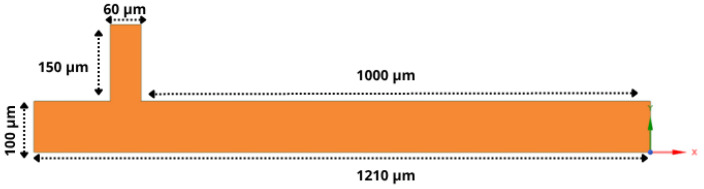
Dimensions of the channels in the T-junction geometry used in the simulations. The inlet channel for the dispersed phase has a width of 60μm, while the inlet channel for the continuous phase has a width of 100μm.

**Figure 4 micromachines-16-00757-f004:**
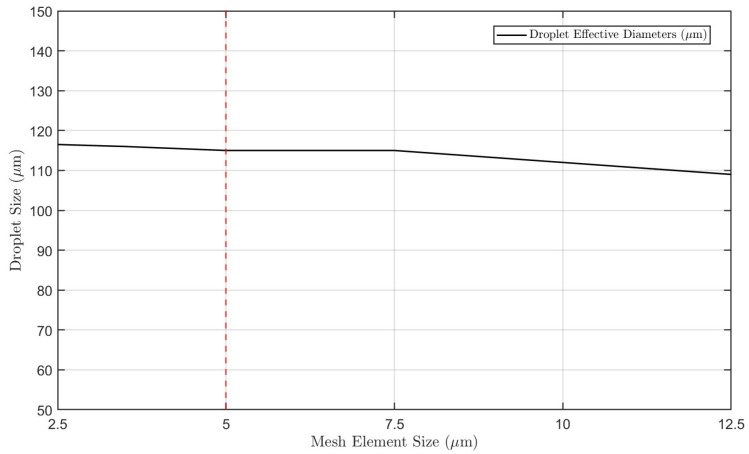
Mesh independence study. A 5 μm mesh was selected as it offered a good balance between accuracy and computational cost, with no significant differences in the analyzed parameter compared to finer meshes.

**Figure 5 micromachines-16-00757-f005:**
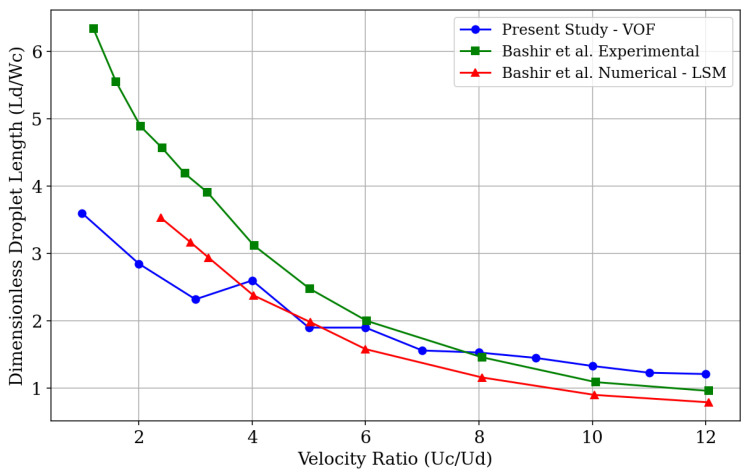
Comparison of droplet length for flow velocity ratios between the present study and the validation model (Bashir et al., 2011 [11]), which includes experimental and numerical data.

**Figure 6 micromachines-16-00757-f006:**
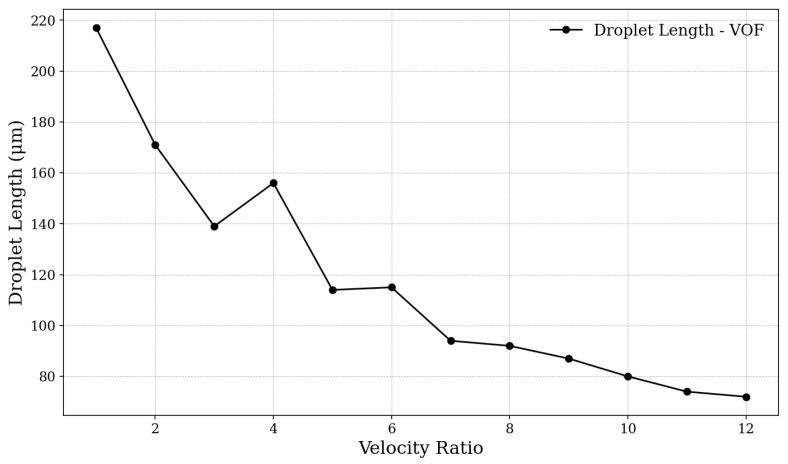
Dimensionless droplet length as a function of the velocity ratio ($U_{c}/U_{d}$). A general decreasing trend in droplet size is observed as $U_{c}/U_{d}$ increases, except near $U_{c}/U_{d}\approx 4$, where an irregular pattern emerges with an unexpected increase in droplet length.

**Figure 7 micromachines-16-00757-f007:**
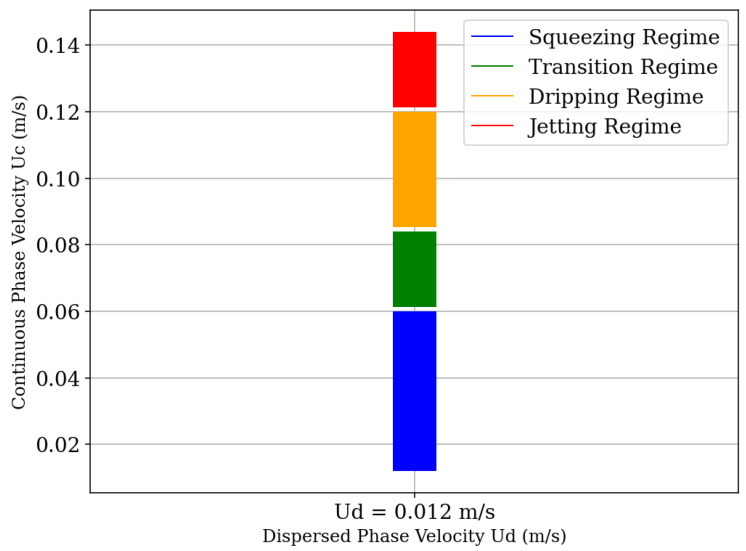
Phase diagram illustrating the different droplet formation regimes as a function of the continuous phase velocity ($U_{c}$) for a fixed dispersed phase velocity ($U_{d}=0.012\;m/s$). The regimes are defined over the following intervals of $U_{c}$: squeezing regime ($0.015\leq U_{c}<0.060\;m/s$), transition regime ($0.060\leq U_{c}<0.085\;m/s$), dripping regime ($0.085\leq U_{c}<0.115\;m/s$), and jetting regime ($0.115\leq U_{c}\leq 0.140\;m/s$).

**Figure 8 micromachines-16-00757-f008:**
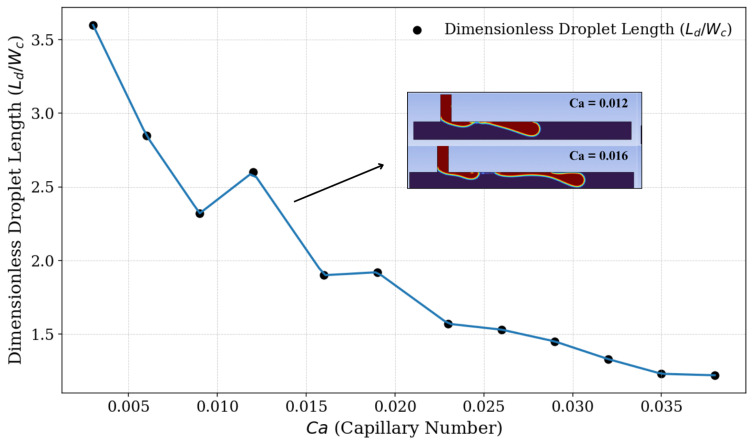
Dimensionless droplet length ($L_{d}/W_{c}$) as a function of the capillary number (*Ca*). The results indicate a general decrease in droplet length as the capillary number increases. A slight increase in droplet length is observed around Ca≈0.012, representing a local deviation from the overall decreasing behavior.

**Figure 9 micromachines-16-00757-f009:**
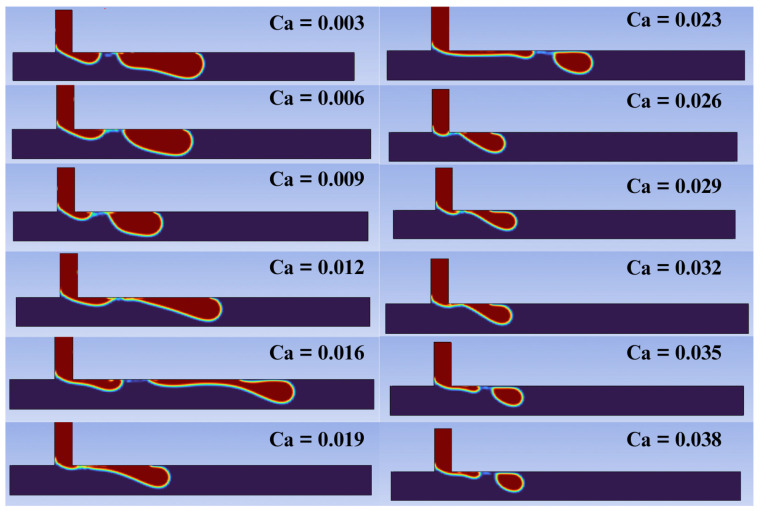
Droplet break up for different capillary numbers. The breakup moments of the dispersed phase at the junction were captured under different capillary number (Ca) conditions. These observations provide valuable insights into the droplet formation regimes as a function of the capillary number.

**Figure 10 micromachines-16-00757-f010:**
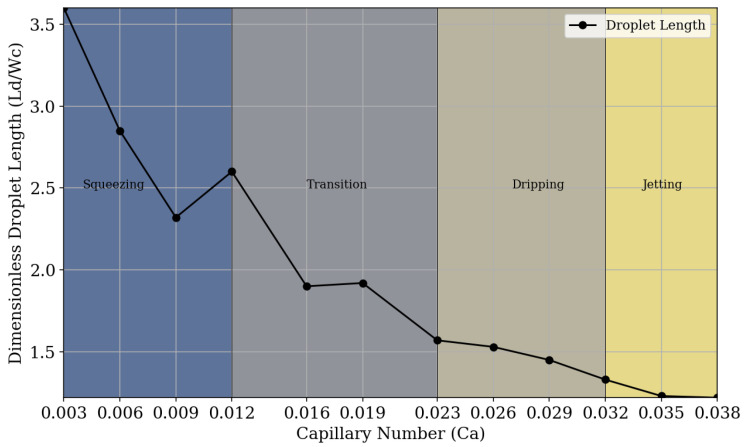
Phase diagram of fluid regimes depending on the capillary number. The squeezing regime is observed in the range of 0.003<Ca<0.012. A transition region, associated with unstable droplet formation, occurs in the range of 0.012<Ca<0.023. The dripping regime appears within 0.023<Ca<0.032, and for Ca>0.032, the system enters the jetting regime.

**Table 1 micromachines-16-00757-t001:** Physical properties of the fluids used in the simulations.

Fluid	Density (ρ, kg/$m^{3}$)	Viscosity (η, mPa · s)
Water	1000	1.0
n-Dodecane Oil	750	1.34

**Table 2 micromachines-16-00757-t002:** Qualitative comparison between the VOF method and LSM based on the present study and the numerical validation by Bashir et al. [11].

Criterion	VOF (Present Study)	LSM (Bashir et al.) [11]
Interface Representation	Volume fraction function	Distance function
Mass Conservation	High (better conservation)	Limited (mass loss observed)
Droplet Size Prediction at Low $U_{c}/U_{d}$	Underestimation compared to experimental data	Closer to experimental data
Droplet Size Prediction at High $U_{c}/U_{d}$	Improved agreement with experiments	Good agreement with experiments
Identification of Regime Transitions	Effective in identifying transition and instability regions	Less detailed in transition mapping
Suitability for Microfluidics	Suitable and efficient for analyzing physical parameters	Higher accuracy in interface shape

## Data Availability

The datasets used and analyzed during the current study are available from the corresponding author on reasonable request.

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
