# Peer review of "Volume of Fluid (VOF) Method as a Suitable Method for Studying Droplet Formation in a Microchannel"

_micromachines, 2025, doi:10.3390/mi16070757_

Round 1
Reviewer 1 Report
Comments and Suggestions for Authors
The manuscript studied the applicability of the Volume of fluid (VOF) method in analyzing droplet generation within microchannels. The authors compared the numerical approaches, VOF and Level Set Method (LSM), highlighting VOF's effectiveness in revealing non-uniformities in droplet sizes and transition regions between flow regimes. This provides valuable insights into microfluidic instabilities and state transitions.
However, several questions should be addressed before it can be accepted for publication:
- The authors should, for instance, compile a comprehensive table comparing the detailed performance of the two numerical methods.
- The manuscript compared VOF and LSM; additional experimental data could strengthen the credibility of the findings. Are there any discrepancies observed when comparing numerical results with experimental observations?
- The influence of mesh resolution on interface curvature estimation should be discussed. Does the choice of grid size impact the accuracy of droplet size predictions?
- The choice of interfacial tension and viscosity should be critically evaluated, particularly in cases where these properties vary significantly. The value should reflect the application of droplet microfluidics in the real world.
Author Response
Comment 1:
1. The authors should, for instance, compile a comprehensive table comparing the detailed performance of the two numerical methods.
Response 1:
Thank you for pointing this out. We agree with this comment. In this context, we included in the manuscript (page 18) a comparative table between the two methods addressed (VOF and LSM) to quantitatively highlight the main differences and implementation approaches of these methods in the study of microfluidics. We discussed criteria such as interface representation, mass conservation, droplet size prediction in relation to experimental data, identification of instability regimes, and applicability in microfluidics. Finally, we evaluated the performance of the methods based on the present study and the results from the validation study (Bashir et al., 2011).
Comment 2:
2. The manuscript compared VOF and LSM; additional experimental data could strengthen the credibility of the findings. Are there any discrepancies observed when comparing numerical results with experimental observations?
Response 2:
Thank you for the comment. We understand the importance of the relationship between numerical and experimental data for a broader understanding of the simulation limits. Since our goal was to conduct a theoretical study focused on the VOF method, the comparison was predominantly numerical, with the understanding that, for the reference model, the experimental observations had already been carried out for validation purposes. Thus, we used the work of Bashir et al. (2011) as the validation reference for our model, aiming to understand which differences arising from the application of different methods could be observed.
Comment 3:
3. The influence of mesh resolution on interface curvature estimation should be discussed. Does the choice of grid size impact the accuracy of droplet size predictions?
Response 3:
Thank you for your comment. The mesh study is presented on page 09 of the manuscript. We addressed the comparison of the effects of different mesh element sizes on the droplet diameter, aiming to understand how this physical parameter varies with mesh refinement and, therefore, to identify the element size at which the variation becomes minimized. To clarify how the geometric mesh affects interface resolution and to promote understanding of the need to balance accuracy and computational cost, we have outlined these points more clearly in the manuscript (page 08).
Response 4:
4. The choice of interfacial tension and viscosity should be critically evaluated, particularly in cases where these properties vary significantly. The value should reflect the application of droplet microfluidics in the real world.
Response 4:
Thank you for your comment. The choice of interfacial tension and fluid viscosity values was made according to the values used in the study that served as numerical validation for the present work. Since the main objective is to understand the suitability of the employed method based on the validation of another numerical study, the values of certain physical parameters (such as those described) were kept the same to avoid altering the nature of the results regarding the comparison between different numerical models.
Reviewer 2 Report
Comments and Suggestions for Authors
The manuscript reported quantitative comparison between VOF and level set method for simulating droplet generation in a microfluidic device. Specifically, based on previous studies which reported experimental results and gave numerical simulation with level set method, authors conducted numerical simulation commercial S/W which adopted VOF in simulating droplet generation. According to the quantitative results, both methods (i.e., VOF and level set method) exhibited consistent results on the effect of fluid velocity as well as capillary numbers. However, the following issues should be improved against publication of the current manuscript as,
- According to the introduction, the manuscript aimed at quantitative comparison between VOF and level set method. Commercial S/W (ANYSYS FLUENT) was adopted to conduct numerical simulation as VOF algorithm. With regard to level set method, instead of numerical simulation, authors used experimental results and numerical simulation obtained by the level set method. Herein, what is originality of the manuscript and authors’ contribution?
- In all figures, figure caption was much short. Please add much information for understanding each figure sufficiently, even without referring to the main text.
- In the Figure 2, numerical value decreased along X-axis. Additionally, Y-axis was positioned in right side rather than left side. It was abnormal. Please correct it.
- As figure 4 included Figure 5 and Figure 6, should authors remove Figure 5 and Figure 6. Otherwise, Figure 4 should be removed.
- In the Figure 7, it would be better to include image on droplet generation. That is, numerical simulation results should be included for understanding droplet generation, which had been influenced by fluid velocity or capillary number. Please add suitable images on droplet generation.
- As an ultimate aim of the manuscript, authors should discuss quantitative comparison between VOF and level set method as one figure.
- In the manuscript, there was no information on a microfluidic channel in detail. For example, looking at the Figure 2, the author did not find out channel information (i.e., width, depth, and length).
- Please correct many typos in text as well as math formula through manuscript.
Author Response
Comment 1:
1. According to the introduction, the manuscript aimed at quantitative comparison between VOF and level set method. Commercial S/W (ANSYS FLUENT) was adopted to conduct numerical simulation as VOF algorithm. With regard to level set method, instead of numerical simulation, authors used experimental results and numerical simulation obtained by the level set method. Herein, what is originality of the manuscript and authors’ contribution?
Response 1:
While previous studies have often favored the Level Set Method (LSM) for its accuracy and mass conservation capabilities, we provide a novel perspective by evaluating the performance and suitability of the VOF method in capturing flow instabilities and non-uniform droplet behaviors within microfluidic regimes. We contribute a quantitative comparison between VOF-based simulations (conducted in ANSYS Fluent) and results from a previously validated numerical-experimental study that employed the LSM. This comparative approach allows us to critically assess the applicability of the VOF method in microfluidics, especially under unstable flow conditions, and to advance the understanding of its strengths and limitations relative to LSM in simulating complex droplet dynamics.
Comment 2:
2. In all figures, figure caption was much short. Please add much information for understanding each figure sufficiently, even without referring to the main text.
Response 2:
Thank you for pointing this out. We have added a more detailed and appropriate caption to the figures, capable of provifing the necessary information.
Comment 3:
3. In the Figure 2, numerical value decreased along X-axis. Additionally, Y-axis was positioned in right side rather than left side. It was abnormal. Please correct it.
Response 3:
Thank you for your comment. We have made the suggested changes, which can be seen on page 09.
Comment 4:
4. As figure 4 included Figure 5 and Figure 6, should authors remove Figure 5 and Figure 6. Otherwise, Figure 4 should be removed.
Response 4:
Thank you for pointing this out. We removed one of the figures that presented ambiguity. We kept the other graphs due to the different approaches and analysis that we aimed to provide. Figure 5 provides a comparative view of the numerical methods and the experimental results, while Figure 6 seeks to outline characteristics of the method used in the study, highlighting the behavior of the droplet size prediction and the regions of instability.
Comment 5:
5. In the Figure 7, it would be better to include image on droplet generation. That is, numerical simulation results should be included for understanding droplet generation, which had been influenced by fluid velocity or capillary number. Please add suitable images on droplet generation.
Response 5:
Thank you for your comment. We partially agree with the suggestion. Indeed, the addition of droplet generation images enhances the understanding of the formation regimes. However, the figure explicitly highlights the dependence of these regimes on the velocity values. Images of droplet formation (particularly the breakup of the dispersed phase) have been included in Figure 09, where they are linked to the capillary number, which is, in turn, directly related to the flow velocities. It is also through the visualization of Figure 09 that the different formation regimes can be clearly identified.
Comment 6:
6. As an ultimate aim of the manuscript, authors should discuss quantitative comparison between VOF and level set method as one figure.
Response 6:
Thank you for pointing this out. We agree with this comment. In this context, we included in the manuscript (page 18) a comparative table between the two methods addressed (VOF and LSM) to quantitatively highlight the main differences and implementation approaches of these methods in the study of microfluidics. We discussed criteria such as interface representation, mass conservation, droplet size prediction in relation to experimental data, identification of instability regimes, and applicability in microfluidics. Finally, we evaluated the performance of the methods based on the present study and the results from the validation study (Bashir et al., 2011).
Comment 7:
7. In the manuscript, there was no information on a microfluidic channel in detail. For example, looking at the Figure 2, the author did not find out channel information (i.e., width, depth, and length).
Response 7:
Thank you for pointing this out. We have added a new figure that makes the dimensions of the channels of the geometry studied explicit (page 08).
Comment 8:
8. Please correct many typos in text as well as math formula through manuscript.
Response 8:
Typos in the text as well as in the mathematical formulas throughout the manuscript have been corrected.
Round 2
Reviewer 1 Report
Comments and Suggestions for Authors
The authors can check the manuscript. The cited references appear as "?" but not numbers.
Reviewer 2 Report
Comments and Suggestions for Authors
As some issues which the reviewer raised were discussed sufficiently, the reviewer
suggests the publication of the manuscript as current from.